# CAN LARGE LANGUAGE MODELS MODEL PROGRAMS FORMALLY?

## ABSTRACT

In the digital age, ensuring the correctness, safety, and reliability of software through formal verification is paramount, particularly as software increasingly underpins critical infrastructure. Formal verification, split into theorem proving and model checking, provides a feasible and reliable path. Unlike theorem proving, which yields notable advances, model checking has been less focused due to the difficulty of automatic program modeling. To fill this gap, we introduce MODEL-BENCH, a benchmark and an accompanying pipeline for evaluating and improving LLMs' program modeling capability by modeling Python programs into verification-ready model checking specifications checkable by its accompanying model checker. MODEL-BENCH comprises 400 Python programs derived from three well-known benchmarks (HumanEval, MBPP, and LiveCodeBench). Our extensive experiments reveal significant limitations in LLMs' program modeling and further provide inspiring directions.

## 1 INTRODUCTION

In an era when software defines everything, daily life is mediated by code across healthcare, finance, and critical infrastructure. A single defect can trigger outages, breaches, or safety incidents, making correctness, safety, and reliability the bedrock of a resilient digital society. While software testing can reveal the presence of bugs, it cannot prove their absence; formal verification, when grounded in precise specifications, can provide machine-checkable guarantees. Technically, formal methods split into two main approaches: *theorem proving*, which establishes properties via logical derivations in proof assistants or automated provers, and ***model checking*** (Clarke, 1997), which decides property satisfaction by exhaustively exploring a system's state space against temporal specifications.

Recent progress has concentrated on LLM-assisted theorem proving (*e.g.*, autoformalization (Wu et al., 2022; Jiang et al., 2024), proof generation (Yang et al., 2023), and premise selection and retrieval), yielding notable advances. By contrast, model checking has been less focused, largely due to the automodeling bottleneck: it is difficult to derive accurate and tractable behavioral models from programs automatically. Though there are a few attempts to model formal properties from requirements (Cao et al., 2025a), modeling formal models for programs has rarely been explored.

However, automatically constructing such models from code is technically challenging and underexplored. Dynamic languages like Python exhibit rich runtime behavior (*e.g.*, mutable aliasing, higher-order functions, third-party libraries, async/await) that must be abstracted to a finite but faithful state space. Useful models are expected to keep a soundness and precision trade-off: too concrete and model checkers do not scale; too abstract and properties become vacuous or unsound.

This gap motivates our work, MODEL-BENCH, a benchmark and an accompanying pipeline for evaluating and improving LLMs' program modeling capability by modeling Python programs into verification-ready TLA+ (Temporal Logic of Actions, a formal language for model checking) (Lamport, 2002) specifications checkable by its accompanying model checker TLC (Yu et al., 1999). MODEL-BENCH comprises 400 Python programs derived from three well-known benchmarks (*i.e.* HumanEval (Chen et al., 2021), MBPP (Austin et al., 2021), and LiveCodeBench (Jain et al., 2024)) by normalization, simplification, and rewrite. The benchmark covers progressively difficult settings, from easy to medium, and then to hard. These programs are covered by a total of 1,639 test cases, ensuring a rigorous evaluation.

Our extensive experiments reveal significant limitations in LLMs' program modeling: only 66.25% runnable and 49.55% state similarity under in-context learning at best. We also propose a code transformation approach to facilitate LLMs modeling and yield promising complementary improvements. Finally, we showed that the modeling difficulty is not reliably correlated with algorithmic difficulty but with nested loops and data-structure complexity. Our contribution includes:

- *Significance.* We proposed MODEL-BENCH, a benchmark and an accompanying pipeline for evaluating and improving LLMs' program modeling capability by modeling Python programs into verification-ready TLA+
- *Novelty.* Besides introducing MODEL-BENCH, we also demonstrate a way to improve the LLMs' program modeling capability via code transformation.
- *Evaluation.* We conduct extensive experiments that yield several instructive findings. Our analysis of bad cases also provides directions for future improvement.

## 2 RELATED WORK

**Automated Formal Verification** While there exist various approaches and techniques for automated formal verification that generates program specifications from natural language (Cosler et al., 2023; Zhai et al., 2020; Giannakopoulou et al., 2020), our MODEL-BENCH primarily focuses on specification generation based on the programming language. In recent years, there also has been a growing interest in applying LLMs to assist program verification (Lin et al., 2024; Ling et al., 2023; Wang et al., 2023; Huang et al., 2024; Jiang et al., 2022). These works focus on using LLMs for theorem proving or domain-specific modeling. For example, Zhou (Zhou, 2025) introduces a two-stage proof generation method that combines LLMs with Retrieval-Augmented Generation. Additionally, frameworks such as CryptoFormalEval (Curaba et al., 2024), AVRE (Yang & Wang, 2024), and Mao et al. (Mao et al., 2025) have designed specialized automated modeling and verification architectures for specific domains, such as cryptographic protocols and 5G communication protocols. Our MODEL-BENCH represents the first LLM-based, general-purpose research effort focused on generating model specifications directly from source code.

**Formal Verification Benchmarks** The formal specification benchmarks offer a standard, well-defined set of problems, providing a shared challenge that helps build a community of practice among researchers. **For formal theorem proving**, a recent survey (Li et al., 2024) summarized the existing datasets. NL-PS (Ferreira & Freitas, 2020) first builds a natural language premise selection dataset source from ProofWiki. Similarly, NaturalProofs (Welleck et al., 2021) further incorporates data from Stacks and textbooks, resulting in a dataset with roughly 25k examples. Adapted from it, NaturalProofs-Gen (Welleck et al., 2022) contains around 14.5k theorems for informal proof generation. Moreover, FM-bench (Cao et al., 2025b) constructed 18k high-quality instruction-response pairs across five mainstream formal specification languages (Coq, Lean4, Dafny, ACSL, and TLA+). **For model checking**, there are few benchmarks and datasets. FM-bench (Cao et al., 2025b) includes benchmarks in TLA+ that evaluates the LLMs' ability to turn informal language to formal specification. So MODEL-BENCH takes the first step in auto modeling from programs with LLMs.

## 3 BENCHMARK CONSTRUCTION

### 3.1 DATA PROCESSING

The workflow of data processing for MODEL-BENCH is illustrated in Figure 1.

**Data Sources** MODEL-BENCH originates from three Python benchmarks for LLM evaluation: HumanEval, MBPP and LiveCodeBench. We select them because they are the most widely known and commonly used function-level benchmarks. All of them provide problem-wise testcases, allowing MODEL-BENCH to validate the correctness of the generated TLA+ models.

**Fetching and Normalization** The workflow begins with the data collection, where we download all three raw datasets from Hugging Face(HumanEval, MBPP, LiveCodeBench) (hug, 2016). The code solutions in Python and the corresponding test cases are extracted from the raw datasets. In order to standardize our benchmark and ensure consistent evaluation, each problem is de-duplicated, normalized and combined into an isolated Python file with a function and test assertions.

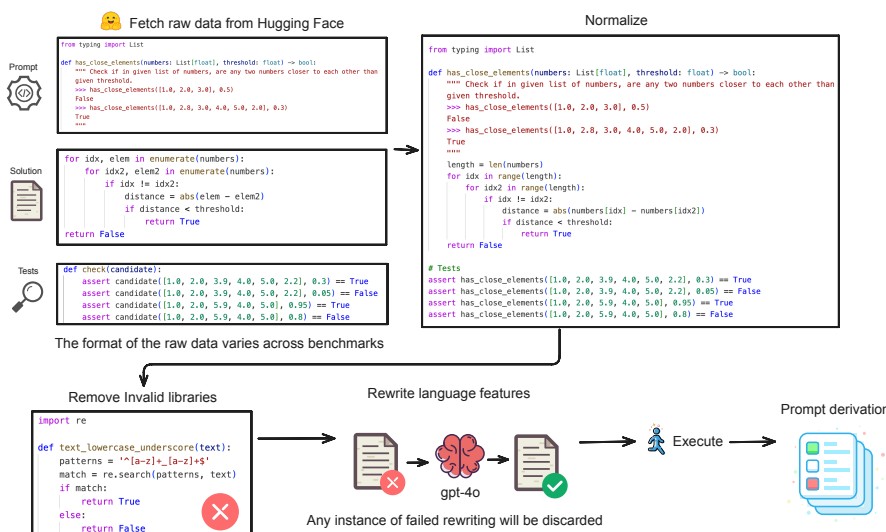

Figure 1: Overview of Data Processing

**Simplification and rewrite** Our focus is on how LLMs abstract and model the core program logic rather than translating every line of code with complex high-level syntax. However, Python is a programming language equipped with rich built-in libraries and modern language features that cannot be easily expressed in modeling languages like TLA+. For built-in libraries, we eliminate all Python code that imports libraries other than `typing` and `math`. Having LLMs continuously generate code for all complex dependencies and their nested dependencies would deviate from our research focus. We retain `typing` because its usage does not affect the code logic in any way and these dependencies can be completely ignored during modeling. We also keep `math` since it contains convenient mathematical functions that are typically simple and commonly used.

For language features, we identify those that require treatment: multiple function declarations, recursion, list comprehension, slice operations, classes, lambda expressions, and generators. Instead of directly discarding these programs, we perform preprocessing and instructing LLMs to equivalently rewrite all programs with these features using the prompt shown in Figure 12. Only after multiple rewriting attempts were problems that still fail to run or meet the requirements discarded. Finally, we exclude Python problems involving variables with complex types beyond `None`, `Number`, `String`, and their derived `List`, `Tuple`, `Dict` and `Iter`, as these types are difficult to represent in TLA+. Our statistics show that this filtering only eliminates a negligible portion of Python problems. Subsequently, the Python problem files undergo execution to verify their functionality and accuracy, with all problematic files that fail this verification process being eliminated. The code must also be acceptable and processable by our code transformer.

**Prompt derivation** For each filtered Python code, we derive three variants of the prompts from the same template shown in Figure 15: original code supplemented with two examples, original code without examples, and transformed code (Section 3.2) with two examples. The prompt template contains fixed domain knowledge of TLA+ and instructions of common mistakes (Lu et al., 2024). The examples are TLA+ models manually crafted from Python programs, designed to provide maximum reference value for LLMs.

## 3.2 CODE TRANSFORMATION

One significant distinction between Python (or other modern high-level programming languages) and TLA+ lies in their fundamental execution models. TLA+ models are essentially state machines that explicitly describe all possible program behaviors as a set of flat, discrete events (*actions*), representing program execution as a sequence of state transitions triggered by these actions. Previous research has indicated that LLMs excel more at imitation and pattern recognition rather than com-

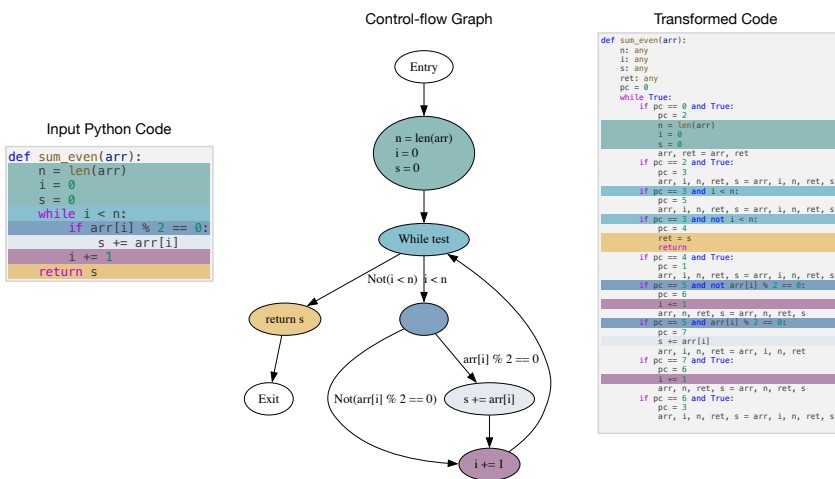

Figure 2: Overview of Code Transformation

Table 1: Data Statistics of MODEL-BENCH

| Source | Origin | Libraries | Language Features | Types | Execution |
|---|---|---|---|---|---|
| HumanEval | 164 | 156 | 139 | 122 | 105 |
| MBPP | 427 | 356 | 334 | 309 | 262 |
| LiveCodeBench | 92 | 73 | 55 | 48 | 33 |
| **Total** | **683** | 595(-12.9%) | 528 (-11.2%) | 479 (-9.2%) | 400 (-16.5%) |

plex reasoning. To investigate the effectiveness of this approach, we *lower* Python programs into a representation more closely aligned with TLA+ models.

The overview of our code transformation is shown in Figure 2. It starts with converting Python programs into control-flow graphs(CFG), where node represent basic blocks, and edges denote either conditional or unconditional jumps between blocks. Each basic block contains a sequence of consecutively executed instructions, which naturally corresponds to actions in TLA+ specifications. This structural similarity enables us to bridge the gap between the two representations while preserving the behavior of the original program.

The transformation process involves several key steps: (1) CFG construction. We construct a CFG from the Python program's abstract syntax tree. Control flow statements (if/else, while/for, break/continue) are identified to partition the code into basic blocks, while recording transition conditions between blocks. Then we assign unique numerical identifiers to each node in the CFG. (2) Code generation. We generate the transformed code following a state machine pattern. All variables are declared at the beginning, followed by introducing a pc variable to track the current state. The main structure is a while loop, containing if statements for each node, controlled by pc and transition conditions. (3) Finally, we also lower Python strings to number arrays based on characters' ASCII values because the strings in TLA+ are immutable.

### 3.3 DATA STATISTICS

The data statistics of MODEL-BENCH are shown in Table 1. In particular, it presents a detailed breakdown of the number of reserved problems in each stage of data processing. Our filtering process resulted in the elimination of approximately 41.4% of the initial dataset, comprising 26.2% from HumanEval, 65.5% from MBPP, and 8.3% from LiveCodeBench. The sequential filtering stages exhibited rates of 12.9%, 11.2%, 9.2%, and 16.5% respectively, representing reasonable attrition levels for maintaining data quality.

## 4 EXPERIMENT DESIGN

We instruct LLMs to model the program across three prompt settings, conducting three sampling trials for each. In these trials, LLMs self-correct via error feedback and iterative multi-turn chats. We also perform post-processing on the models generated by LLMs (Section A.1). The processed models are evaluated using TLC (Yu et al., 1999) and two metrics (Section 4.1.1).

### 4.1 EVALUATION DESIGN

**Evaluation Preparation**. We employ GPT-4o to generate a model for each Python program. Through manual verification and refinement, we obtain oracle models. These models serve as the ground truth for evaluating the similarity (defined below) of models generated by LLMs.

#### 4.1.1 EVALUATION METRICS

**Runnable@$k$**: derived from Pass@$k$ (Chen et al., 2021), a popular metric in LLM evaluation:

$$Runnable@k = \mathbb{E}\left[1 - \binom{n-c}{k} / \binom{n}{k}\right] \tag{1}$$

Here, we define it as the proportion of models that TLC checks without failures at least once within $k$ generated models. For each problem, $n$ solutions are sampled from an LLM, and $c$ of $n$ solutions are correct. Considering the time and cost, we set $n$ to 3 and $k$ to 1, 2, 3 for each model sampling.

**Similarity**: Previous research has demonstrated that LLMs may not strictly adhere to prompts (Liu et al., 2023). To verify whether the LLM-generated models align with the original programs rather than being complete rewrites, we introduce a similarity metric. This metric measures the proportion of identical states between two state transition sequences, formally defined as:

$$Similarity(M_o, M_g) = \frac{|\{s | s \in States(M_o) \cap States(M_g)\}|}{|States(M_o)|} \tag{2}$$

where $M_o$ denotes oracle models, $M_g$ denotes LLMs-generated models, and $States(P)$ denotes the set of all states observed during the execution of model $M$ with TKC, formally defined as the union of all states at each time step $t$:

$$States(M) = \bigcup_t State(M, t) \tag{3}$$

Each state at time $t$ is defined as the set of variable-value pairs:

$$State(M, t) = \{(v, val(v, t)) \mid v \in Vars(M)\} \tag{4}$$

Here, $Vars(M)$ represents the set of variables in model $M$, $val(v, t)$ denotes the value of variable $v$ at step $t$, and $States(M)$ is the set of all states in models $M$'s execution trace.

Two states $State_1$ and $State_2$ are considered **sufficiently similar**, if and only if the proportion of variable values in $State_g$ that also exist in $State_o$ is greater than or equal to a threshold $\theta \in [0, 1]$. Formally,

$$\frac{|\{(v_g, val_g(v_g, t_g)) \in State_g \mid \exists (v_o, val_o(v_o, t_o)) \in State_o, val_g(v_g, t_g) = val_o(v_o, t_o)\}|}{|State_g|} \geq \theta \tag{5}$$

We set the threshold $\theta$ to 1.0 to ensure that all variable-value pairs in the state of the generate model must be present in the state of the oracle model, i.e., no discrepancies or noise. Note that higher Runnable@$k$ doesn't mean higher similarity. This metric represents a compromise, given the challenge of fully assessing whether the execution process and semantics of the program and the model are entirely aligned.

## 5 EVALUATION

We use nucleus sampling (Holtzman et al., 2019) in line with recent works (Cao et al., 2024b; Du et al., 2023; Cao et al., 2024a; Yu et al., 2024). All solution samples are randomly generated

Table 2: Overall Results on MODEL-BENCH

| Model | Runnable@1 (%) | Runnable@2 (%) | Runnable@3 (%) | Avg Similarity (%) | Avg Fixes |
|---|---|---|---|---|---|
| **Original Code / Few-shot** | | | | | |
| DeepSeek-V3 | 51.75 | 61.33 | 66.25 | 49.55 | 0.78 |
| DeepSeek-V2.5 | 44.33 | 53.50 | 57.75 | 46.17 | 0.78 |
| Qwen3-32B | 39.50 | 53.08 | 60.75 | 52.03 | 1.07 |
| Qwen3-14B | 29.75 | 41.50 | 49.25 | 43.60 | 1.23 |
| DeepSeek-R1-Distill-Qwen-32B | 21.00 | 30.42 | 36.50 | 30.15 | 1.38 |
| Qwen3-8B | 16.00 | 24.75 | 31.00 | 22.84 | 1.90 |
| Gemma-3-12b-it | 7.92 | 12.17 | 15.00 | 21.14 | 1.40 |
| Llama-3.1-8B-Instruct | 4.33 | 7.33 | 10.00 | 5.39 | 2.87 |
| **Average** | **26.82** | **35.51** | **40.81** | **33.86** | **1.43** |
| **Original Code / Zero-shot** | | | | | |
| DeepSeek-V3 | 37.92 | 52.92 | 61.00 | 4.38 | 2.29 |
| DeepSeek-V2.5 | 12.75 | 21.17 | 27.25 | 3.08 | 2.68 |
| Qwen3-32B | 18.08 | 29.75 | 37.75 | 11.90 | 2.49 |
| Qwen3-14B | 1.17 | 1.33 | 1.50 | 0.00 | 4.71 |
| DeepSeek-R1-Distill-Qwen-32B | 1.08 | 1.17 | 1.25 | 0.00 | 5.00 |
| Qwen3-8B | 1.00 | 1.00 | 1.00 | 0.00 | 5.00 |
| Gemma-3-12b-it | 1.00 | 1.00 | 1.00 | 0.00 | 5.00 |
| Llama-3.1-8B-Instruct | 1.00 | 1.00 | 1.00 | 0.00 | 5.00 |
| **Average** | **9.25** | **13.67** | **16.47** | **2.42** | **4.02** |
| **Transformed Code / Few-shot** | | | | | |
| DeepSeek-V3 | 43.50 | 51.75 | 56.00 | 68.54 | 0.71 |
| DeepSeek-V2.5 | 39.92 | 48.17 | 52.25 | 61.55 | 0.84 |
| Qwen3-32B | 33.42 | 45.42 | 53.25 | 62.64 | 0.68 |
| Qwen3-14B | 26.08 | 37.08 | 44.25 | 57.67 | 1.48 |
| DeepSeek-R1-Distill-Qwen-32B | 24.00 | 35.08 | 41.25 | 52.95 | 1.69 |
| Qwen3-8B | 16.83 | 25.42 | 31.50 | 50.19 | 1.75 |
| Gemma-3-12b-it | 7.42 | 10.67 | 12.50 | 39.31 | 2.00 |
| Llama-3.1-8B-Instruct | 5.50 | 8.83 | 11.75 | 40.71 | 2.47 |
| **Average** | **24.58** | **32.80** | **37.84** | **54.20** | **1.45** |

with a temperature of 0.7 (Wen et al., 2024), which is the default temperature of ChatGPT. Due to computational constraints, only the Gemma and Llama models are run on our local server equipped with two NVIDIA RTX 6000 Ada GPUs (each with 48GB of graphic memory). The remaining models are executed through the SiliconFlow API sil (2023).

The research questions (RQs) were designed as follows:

- ***RQ1. Overall Performance.*** We first show the overall performance of the studied LLMs on MODEL-BENCH. We use three sets of prompts to generate modelings for all Python code. The comprehensive results are displayed with multiple metrics.

- ***RQ2. Effectiveness of Code Transformation.*** The transformed code (Section 3.2) more closely resembles the form of a TLA+ model. We thus explore how this approach affects different LLMs.

- ***RQ3. Impact of Source Code Syntactic Complexity.*** Research indicated that the accuracy of code generated by LLMs is negatively correlated with code complexity (Sepidband et al., 2025). So we aim to explore the relationship between the performance of LLMs in automated modeling and the complexity of the code involved.

- ***RQ4. Bad Case Analysis.*** We analyze bad cases with syntactic or semantic errors due to various issues and identify the limitations of LLMs in TLA+ automated modelings.

### 5.1 RQ1: OVERALL PERFORMANCE

The overall performance of the studied LLMs on MODEL-BENCH is shown in Table 2, which lists various metrics for TLA+ models generated by each studied model under three prompt settings (Section 3.1). Metrics include TLC Runnable@$k$, average state similarity between the TLA+ models and oracle models, as well as the average number of fixes under Runnable@1 (Section 4.1.1). To

better visualize the results, we use darker background colors to indicate larger values. Only models that pass TLC verification have the opportunity to calculate state similarity with oracle models.

Generally, across all prompt settings, DeepSeek-V3 demonstrates the best performance, followed by DeepSeek-V2.5 and Qwen3-32B, achieving Runnable@1 of 51.75%, 44.33%, and 39.50%, respectively. These three models also have the top-3 highest average similarities and lowest average fix counts, indicating they can generate models that pass verification within fewer iterations.

> **Finding 1:** The Top-3 performing LLMs are DeepSeek-V3, DeepSeek-V2.5, and Qwen3-32B among the studied LLMs, achieving Runnable@1 rates of 51.75%, 44.33%, and 39.50%, respectively. Their performance rankings remain consistent across all three prompt settings.

Comparing few-shot and zero-shot results, all models demonstrate significantly better performance with few-shot prompt than with zero-shot, averagely improving 17.57% (26.82% - 9.25%) in Runnable@1 and 31.44%(33.86% - 2.42%) in similarity. Particularly, DeepSeek-V2.5 showed a 31.58% improvement (44.33% - 12.75%) in Runnable@1. With few-shot prompt, the three lowest-ranking models achieve a breakthrough from near 0 Runnable@1, with Llama-3.1-8B-Instruct improving to 4.33%, Gemma-3-12b-it reaching 7.92%, and Qwen3-8B achieving 16.00%. For higher-ranked models, few-shot prompt also reduces the average number of fix attempts and increases the average state similarity. For example, DeepSeek-V2.5's average fix attempts decrease by 1.90 (from 2.68 to 0.78), while its average state similarity improves by 46.47% (from 3.08% to 49.55%).

> **Finding 2:** The enhancement of few-shot learning for automatic modeling tasks is substantial. Notably, with zero-shot, models such as Gemma-3-12b-it, Llama-3.1-8B-Instruct, DeepSeek-R1-Distill-Qwen-32B, Qwen3-14B, and Qwen3-8B all demonstrate a nearly 0 Runnable@1 rate.

In addition, by comparing the Runnable@1 rates of all models, we observe that the automatic modeling task from Python to TLA+ exhibits high discriminability. This finding suggests that when applying this technique in actual industrial production, employing more powerful models often yields significant improvements.

## 5.2 RQ2: EFFECTIVENESS OF CODE TRANSFORMATION

In RQ2, we primarily compare the results of original Python code and transformed Python code in a few-shot setting. By comparing the data in Table 2, we observe that for all models, code transformation leads to a decrease in Runnable@$k$, but significantly improves similarity. For instance, in the case of the DeepSeek-V3 model, similarity increases by 18.99% (68.54% - 49.55%), while Runnable@3 decreases by only 10.25% (66.25% - 56.00%). This indicates that code transformation indeed provides LLMs with references that are easier to follow and translate. We hypothesize that the decline in Runnable@$k$ can be attributed to two main factors: (1) It reduces instances where LLMs bypass TLC by completely restructuring the program. (2) Code transformation increases code length, making the "lost in the middle" (Liu et al., 2023) phenomenon more likely to occur.

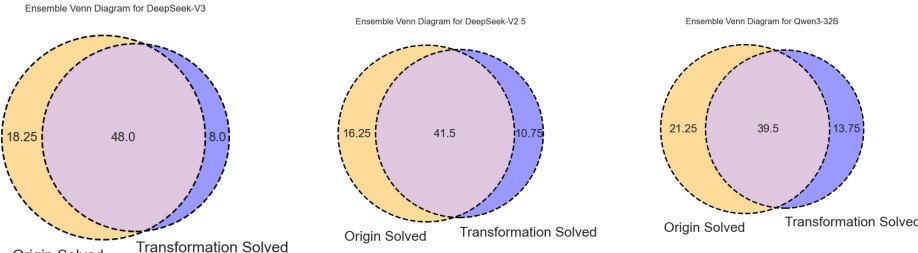

Figure 3: Venn Diagram of Ensemble based on Runnable@3

Figure 3 illustrates the complementary effects of the two prompt settings based on Runnable@3. More results are demonstrated in Figure 7. Assembling the results from both prompt settings proves

to be effective, *e.g.*, for Qwen3-32B, prompting with transformed code yields an additional 13.75% runnable TLA+ models from Python programs, compared that with the original code alone.

> **Finding 3:** Code transformation significantly improves similarity while only leads to a small decrease in Runnable@$k$. It can still serve as a complementary technique for all models when combined with original code prompting, increasing the total number of runnable models.

## 5.3 RQ3: IMPACT OF SOURCE CODE SYNTACTIC COMPLEXITY

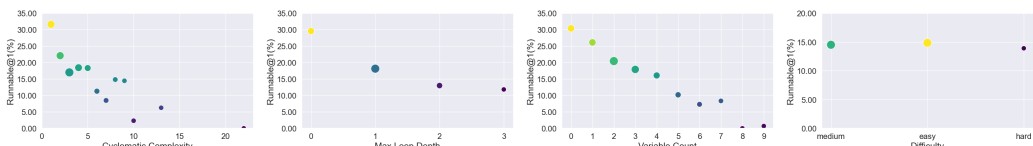

Figure 4: Relationship between Runnable@1 and Code Complexity

This section aims to explore the relationship between the performance of LLMs in automated modeling and the complexity of the code involved. We calculate cyclomatic complexity with the Radon library which is also used by previous work (Sepidband et al., 2025), max loop depth, and the number of variables for each original Python code in advance. We group all problems according to these metrics separately and calculate the proportion of samples in each group that successfully passed the TLC check based on the Runnable@1 results across all models. Figure 4 illustrates the relationship between these complexity metrics and modeling success rates.

We also collect difficulty ratings for Python programs from LiveCodeBench, where these ratings correspond to the difficulty of the problems rather than the complexity of the code. Figure 4 shows the Runnable@$k$ of all models across the three original problem difficulty distributions.

> **Finding 4:** Python programs with higher syntactic complexity, i.e., those exhibiting higher cyclomatic complexity, larger loop depth, and a greater number of variables, demonstrate lower Runnable@$k$ as well as similarity when automatically translated into TLA+ models using LLMs.

## 5.4 RQ4: BAD CASE ANALYSIS

This section discusses cases that cannot pass TLC verification, including scenarios where the generated TLA+ model contains compilation errors, runtime errors, or fails to satisfy assertion properties. Since TLC categorizes errors in a coarse-grained manner via exit codes, we can only roughly determine the cause through manual judgment. These kind of failures also demonstrates that the difference between Python and TLA+ makes automated modeling and code translation hard. More bad cases are demonstrated in Section A.3.2

### 5.4.1 COMPILATION ERROR

Compilation errors indicate that, as a result of insufficient training data or inadequate contextual information, LLMs fail to produce TLA+ code that is syntactically or semantically correct.

**Compilation Error 1: Unknown Operator.** Figure 5 shows an error caused by the absence of `sort` function, which is a built-in function existing in Python. Although we can make LLMs aware of this issue through multiple rounds of chat, such iterative interactions may also continuously introduce new unknown operators. The differences in built-in libraries between the two languages further increase the difficulty of automatic modeling.

### 5.4.2 RUNTIME ERROR

Runtime errors are another type of result stemming from LLMs' lack of attention to differences between languages. Unlike compilation errors, correcting such errors requires LLMs to have a deeper understanding of language specifications.

```python
def sort_third(l):
    """This function takes a list l and returns a list l' such that
    l' is identical to l in the indices that are not divisible by three,
    while its values at the indices that are divisible by three are equal
    to the values of the corresponding indices of l, but sorted.
    >>> sort_third([1, 2, 3])
    [1, 2, 3]
    """
    result = []
    third_indices = []

    for i in range(len(l)):
        if i % 3 == 0:
            third_indices.append(l[i])
            result.append(None)  # Placeholder for sorting later
        else:
            result.append(l[i])

    third_indices.sort()

    third_index = 0
    for i in range(len(result)):
        if result[i] is None:
            result[i] = third_indices[third_index]
            third_index += 1

    return result
```

```
--------------------------- MODULE SortThird ---------------------------
EXTENDS Naturals, Integers, Reals, Sequences, FiniteSets, TLC
VARIABLES l, result, third_indices, i, third_index, ret, initl, pc
vars == <<l, result, third_indices, i, third_index, ret, initl, pc>>

Init == \* ...

SortThird ==
    \* ...
    \/  /\ pc = 4
        /\ pc' = 9
        /\ third_indices' = Sort(third_indices)
        /\ i' = 0
        /\ third_index' = 0
        /\ result' = result
        /\ initl' = initl
        /\ ret' = ret
        /\ l' = l

Next == SortThird

Spec ==
    /\ Init
    /\ [][Next]_vars

Assertion == \* ...
========================================================================
```

Figure 5: Compilation Error 1. Unknown Operator

**Runtime Error 1: One-based Array Index.** Figure 6 shows the error of array index out of bounds. In the Python code, initializing i as 1 leads to the first iteration accessing dp[0]. However, in TLA+, array indices start from 1, which means that dp[0] is an invalid access. One-based arrays are uncommon in most programming languages. Although we explicitly mentioned this in the prompt, LLMs still get lost in the middle of long contexts.

```python
def lengthOfLongestSubsequence(nums, sum):
    n = len(nums)
    dp = []
    for i in range(n + 1):
        dp.append([-1] * (sum + 1))

    for i in range(n + 1):
        dp[i][0] = 0

    for i in range(1, n + 1):
        for j in range(1, sum + 1):
            dp[i][j] = dp[i - 1][j]
            if j >= nums[i - 1] and dp[i - 1][j - nums[i - 1]] != -1:
                dp[i][j] = max(1 + dp[i - 1][j - nums[i - 1]], dp[i][j])

    return dp[n][sum]

assert lengthOfLongestSubsequence([1, 1, 5, 4, 5], 3) == -1
```

```
---------------- MODULE LengthOfLongestSubsequence -----------------
\* ...

Init == \* ...

Next ==
    \/  /\ pc = 6
        /\ i >= n + 1
        /\ pc' = 7
        /\ i' = 1
    \* ...
    \/  /\ pc = 12
        /\ j < sum + 1
        /\ pc' = 14
        /\ j' = j + 1
        /\ dp' = [dp EXCEPT ![i] = [dp[i] EXCEPT ![j+1] = dp[i-1][j+1]]]
        /\ UNCHANGED <<n, initsum, i, sum, nums, initnums, ret>>
========================================================================
```

Figure 6: Runtime Error 1. One-based Array Index

### 5.4.3 ASSERTION ERROR

Assertion errors occur not because LLMs omit the logic in the Python code. This stands in contrast to the two types of errors discussed earlier, as the essence of these errors is rooted in this excessive fidelity to the source code.

**Assertion Error 1: Omission of Function-call.** Figure 11 demonstrates the reason behind the assertion error. The LLMs fail to include the .lower() function-call in Python, which leads to the test cases not passing.

## 6 CONCLUSION

In this paper, we introduce MODEL-BENCH, a benchmark and an accompanying pipeline for evaluating and improving LLMs' program modeling capability by modeling Python programs into verification-ready model checking specifications checkable by its accompanying model checker. MODEL-BENCH comprises 400 Python programs derived from three well-known benchmarks (HumanEval, MBPP, and LiveCodeBench). Our extensive experiments reveal significant limitations in LLMs' program modeling and further provide inspiring directions. We hope MODEL-BENCH could drive progress in automated formal verification, especially for model checking, and encourage the development of more sophisticated reasoning capabilities in future LLMs.

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

## A  APPENDIX

### A.1  POST PROCESSING

Given the limited availability of training data for TLA+ compared to mainstream programming languages, we observe that LLMs tend to make trivial but patterned errors (see below). To ensure more meaningful observations, we conduct post-processing for every generated output. In particular, the post-processing consists of three steps: (1) Import all built-in modules. We automatically incorporate all built-in TLA+ modules through the `EXTENDS` keyword, ensuring access to fundamental operators and definitions required for formal specification. (2) Define null model values. To achieve better correspondence with Python, we introduce `None` and `Null` as model values in the specifications. (3) Complete unchanged variables. We implement comprehensive handling of the `UNCHANGED` variables to ensure that each action's resulting state is complete.

These post-processing steps effectively eliminate common errors that would otherwise impede the validity of our experimental results. This approach allows us to focus on evaluating the substantive aspects of the LLMs' ability to generate formal specifications.

Table 3: Studied Large Language Models

| Model Family | Model | Size | Time |
|---|---|---|---|
| DeepSeek | DeepSeek-V3(Liu et al., 2024b) | 685B | Dec, 2024 |
| DeepSeek | DeepSeek-V2.5(Liu et al., 2024a) | 236B | May, 2024 |
| Qwen | Qwen3-8B(Yang et al., 2025) | 8.19B | May, 2025 |
| Qwen | Qwen3-14B(Yang et al., 2025) | 14.8B | May, 2025 |
| Qwen | Qwen3-32B(Yang et al., 2025) | 32.8B | May, 2025 |
| Qwen | DeepSeek-R1-Distill-Qwen-32B | 32.8B | July, 2024 |
| Gemma | Gemma-3-12B-it(Team et al., 2025) | 12.2B | March, 2025 |
| Llama | Llama-3.1-8B-Instruct(Dubey et al., 2024) | 8.03B | July, 2024 |

## A.2 STUDIED LARGE LANGUAGE MODELS

The studied LLMs are listed in Table 3. We focus on recent LLMs released after 2024. We primarily choose the DeepSeek and Qwen model families, which are the strongest open-source models according to public leaderboards. We specifically include Qwen3-8B, Qwen3-14B, and Qwen3-32B to evaluate how model performance scales with different parameter count within the same family. We incorporate Gemma and Llama to ensure model diversity. Note that we deliberately exclude GPT-series models from OpenAI to ensure fairness, as our later comparison between LLM-generated TLA+ models and oracle models relies on human-corrected outputs based on GPT's generations.

## A.3 EVALUATION SUPPLEMENT

### A.3.1 RQ2: EFFECTIVENESS OF CODE TRANSFORMATION

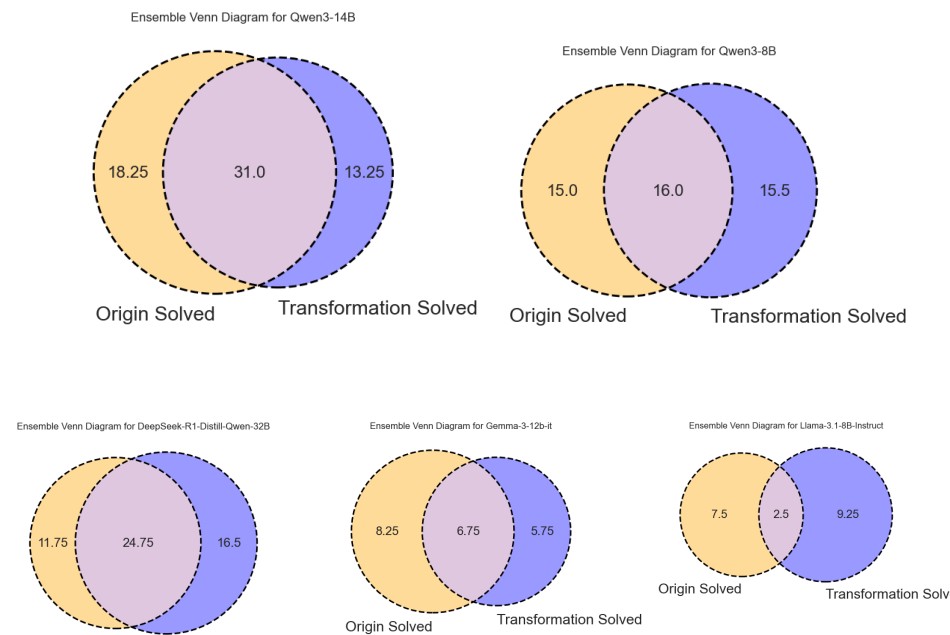

Figure 7: Venn Diagram of Ensemble based on Runnable@3

### A.3.2 RQ4: BAD CASE ANALYSIS

**Compilation Error 2: Unexpected Token.** Figure 8 shows an error caused by invalid string concatenation operator. Python uses + to concatenate two string while TLA+ should use \o. LLMs lack this knowledge, so it generate unexpected tokens as a result.

Figure 8: Compilation Error 2. Unexpected Token

**Runtime Error 2: Indexing String.** Figure 9 illustrates the error that occurs when attempting to index the string `ALPHABET` in TLA+. While in Python, programmers can access individual characters in a string using array-like indexing, this operation is invalid in TLA+, causing runtime errors.

Figure 9: Runtime Error 2. Indexing String

**Assertion Error 2: Constant Loop Count.** Figure 10 illustrates a case where LLMs directly use the constant `MaxLen`, which originally intends to constrain the initial variable search space, as the maximum loop count for traversing an array under arbitrary inputs. The correct way is to use `Len(arr)`.

Figure 10: Assertion Error 2. Constant Loop Count

## A.4    PROMPT DESIGN

```
def check_if_last_char_is_a_letter(txt):              ----------------------- MODULE CheckIfLastCharIsALetter -----------------------
    '''                                               \* ...
    Create a function that returns True if the last character
    of a given string is an alphabetical character and is not    Init == \* ...
    a part of a word, and False otherwise.
    Note: "word" is a group of characters separated by space.   CheckIfLastCharIsALetter ==
                                                          LET IsLetter(c) == /\ Len(c) = 1 /\ 97 <= c[1] /\ c[1] <= 122
    Examples:                                             IN
    check_if_last_char_is_a_letter("apple pie") → False   /\ \/ /\ pc = 0
    check_if_last_char_is_a_letter("apple pi e") → True         /\ check' = check[Len(check)]
    check_if_last_char_is_a_letter("apple pi e ") → False       /\ ret' = IsLetter(check')
    check_if_last_char_is_a_letter("") → False                  /\ pc' = 1
    '''                                                         /\ UNCHANGED <<inittxt, txt>>

    check = txt.split(' ')[-1]
    return True if len(check) == 1 and (97 <= ord(check.lower()) <= 122)   \* ...
else False                                             ================================================================
```

Figure 11: Assertion Error 1. Omission of Function-call

## Prompt for rewriting code of problems

# System Prompt

You are a Python expert. Please refactor the user's Python code into equivalent code following these rules:

1. Avoid using list comprehensions like [x*2 for x in range(5)]. Use traditional for loops instead.

2. Avoid using slicing operations like array[1:4]. Use loops to access elements individually.

3. Avoid using classes with self references like "class Calculator: def add(self, x, y)". Use standalone functions.

4. Avoid using lambda functions like "lambda x: x + 1". Use regular named functions.

5. Avoid using generator expressions like "(x for x in range(5))". Use regular loops and lists.

6. Write single, non-recursive functions instead of recursive ones like "def factorial(n): return n * factorial(n-1)".

Please output the refactored code directly without any additional explanations.

# User Prompt

- Original Python code goes here -

Figure 12: Prompt for Rewriting Code of Problems

## Prompt for fix

The TLA+ specification has the following error:

- error message -

Please fix the specification while keeping the same logic.

Figure 13: Prompt for Fixing

## Config template for running TLC

```
CONSTANTS
- constants -
NONE = NONE
NULL = NULL

SPECIFICATION
Spec

INVARIANT
Assertion

CHECK_DEADLOCK FALSE
```

Figure 14: Config Template for Running TLC

810
811
812
813
814
815
816
817
818
819
820
821
822
823
824
825
826
827
828
829
830
831
832
833
834
835
836
837
838
839
840
841
842
843
844
845
846
847
848
849
850
851
852
853
854
855
856
857
858
859
860
861
862
863

---

**Prompt for generating TLA+ models**

# Role description
As an expert in TLA+, you are good at understanding and writing TLA+.
TLA+ is a formal specification language used for modeling and verifying concurrent and distributed systems.

# Domain knowledge
1. The logical operators supported by TLA+ include:
`/ (and), \/ (or), ~ (not), => (Implication), <=>`
`(Bidirectional implication), TRUE, FALSE, \A (Universal`
`Quantification), \E (Existential Quantification)`

2. The set operators supported by TLA+ include:
`= (Equality), # (not equal), \union (Union), \intersect`
`(Intersection), \in (Membership), \notin (Not in), \subseteq`
`(Subset Equal), \ (Difference).`

3. The temporal operators supported by TLA+ include:
`[] x > 0`
The above code is an example of `[]` (Always). It means that at all times, the value of variable `x` is greater than 0.

`<> x = 0`
The above code is an example of `<>` (Eventually). It means that at some point in time, the value of variable `x` becomes 0.

4. Built-in keywords and operators in TLA+ include:
`MODULE, EXTENDS, CONSTANTS, INSTANCE, VARIABLE, ASSUME,`
`PROVE, INIT, NEXT, ACTION, SPECIFICATION, IF, ELSE, WITH,`
`CASE, THEN, LET, IN, CHOOSE, ENABLED, UNCHANGED, DOMAIN.`

Based on the information and python code with assertions, give a complete TLA+ model code in only one single code block without explanations.
The model should initialize a set of all possible states constrained by max or min `CONSTANT`s instead of fixed inputs.
1. Use `LET` keyword if there's any temporary variable.
2. Each step should define all variables, even though keep them unchange.
3. Since the start index in TLA+ is 1 instead of 0, you may change the corresponding initialization, checks, and assignment.
4. Don't declare parameters with same names as variables or constants.
5. Define arrays like `arr \in [1..MaxLen -> 0..MaxValue]`.
If there are assertions in the code, you should also generate a corresponding `Assertion` == action.

For example:
- `example1` –
- `example2` –

Module Name: – `module_name` –
– `code` –

---

Figure 15: Prompt for Generating TLA+ Models

