# OpenReview forum: "Can Large Language Models Model Programs Formally?"
_ICLR.cc/2026/Conference — ICLR 2026 Conference Withdrawn Submission_

### Official Review · Reviewer_jzTD · 2025-10-19

**Soundness:** 2
**Presentation:** 2
**Contribution:** 2
**Rating:** 2
**Confidence:** 4

**Summary:**

This paper investigates whether large language models can automatically model executable programs into formal specifications suitable for model checking.
The authors propose MODEL-BENCH, a benchmark and pipeline that converts Python programs into TLA+ specifications to test LLMs’ ability to produce verification-ready models.
The dataset contains 400 normalized Python problems (from HumanEval, MBPP, and LiveCodeBench) with 1,639 test cases.
They further design a code-to-state-machine transformation, aligning Python control flow with TLA+ semantics, and evaluate several recent open LLMs (DeepSeek-V3, Qwen3-32B, etc.) under few-shot and zero-shot prompts.

**Strengths:**

- Novel research direction

Shifts LLM formal verification research from theorem proving to automatic model construction, a crucial but underexplored domain.

- Careful benchmark design

Data normalization, feature filtering, and oracle verification provide methodological rigor.

- Insightful diagnostics

Clear categorization of typical failure cases and a quantitative link between syntactic complexity and modeling success.

**Weaknesses:**

- Human-in-the-loop oracle bias

“Ground-truth” TLA+ specs partly rely on GPT-4o plus manual fixes, potentially biasing evaluation.

- Simplified programs

400 cleaned functions miss realistic system-level constructs (I/O, concurrency), reducing ecological validity.

The paper should go to the Dataset and Benchmark area.

**Questions:**

How does the state-similarity metric handle semantically equivalent but syntactically different TLA+ models (e.g., variable renaming, reordering)?
Consider introducing trace-equivalence or property-satisfaction metrics to better reflect true semantic alignment.

Could the pipeline extend to other formalisms (e.g., Alloy, B-Method) or compiled languages (e.g., C/Java)? Discuss design considerations for such adaptation.

Have you explored structured intermediate representations (CFGs, SSA graphs, symbolic traces) as additional supervision to reduce the semantic gap, or reinforcement signals from TLC verification results?

---

### Official Review · Reviewer_XMSg · 2025-10-23

**Soundness:** 2
**Presentation:** 3
**Contribution:** 1
**Rating:** 2
**Confidence:** 5

**Summary:**

In this work, the authors propose a benchmark dataset for evaluating the modeling capability of LLMs for the purpose of model checking. These benchmarks are constructed based on existing Python program datasets and preprocessed through multiple-steps.

**Strengths:**

On the positive side, I enjoy reading the draft for the following reasons.

First, the topic of the study, evaluating the modeling capability of LLMs, is an interesting, important and large overlooked one.

**Weaknesses:**

On the less positive side, the draft can be improved in various aspects.

First of all, the idea of modeling is to establish a sound abstraction based on a level of abstraction which is demanded by the verification task. In other words, modeling is meaningless unless we know what to verify. And this aspect is completely missing from the approach. The authors seem to believe that the model should capture every functional aspect of a given Python program, which is completely the wrong idea. It is not only unnecessary - if I am model checking only whether double-free handles, I only need to model to a point where double-free vulnerabilities are preserved, but also it is impossible - why do you model the execution time of the program?

Second, because of the above, many of the design choices, such as excluding all but trivial Python programs that use no external library or complicated data types, are problematic and unjustified. In fact, I would argue that it makes the benchmark dataset rather limited and useless.

Lastly, the choice of modeling Python program is a problematic one, given that existing model checkers (such as SPIN, TLA, UPPAAL) typically have a reasonably large number of system descriptions and the accompanying models, which can be easily used to construct the benchmark dataset.

The following are a list of detailed comments.

Page 1: “Technically, formal methods split into two main approaches: theorem proving, which establishes properties via logical derivations in proof assistants or automated provers, and model checking …”

Comment: This is rather imprecise given the many other areas of formal methods, such as formal synthesis, and formal specification. Even among formal verification, there are other techniques such as abstract interpretation.

Page 3: “For built-in libraries, we eliminate all Python code that imports libraries other than typing and math. Having LLMs continuously generate code for all complex dependencies and their nested dependencies would deviate from our research focus.”

Comment: I am not sure whether this is a good idea as this would limit the evaluation to the almost trivial Python programs.

Page 3: “Finally, we exclude Python problems involving variables with complex types beyond None, Number, String, and their derived List, Tuple, Dict and Iter, as these types are difficult to represent in TLA+.”

Comment: Again, this seems rather the wrong idea - the idea of modelling is precisely to abstract complex states/operations, and not to model everything precisely.

Page 5: “Through manual verification and refinement, we obtain oracle models. These models serve as the ground truth for evaluating the similarity (defined below) of models generated by LLMs.”

Comment: This is so confusing. How a system or a program should be modeled depends on the properties to be verified and there is never one particular way of modeling. Can you elaborate how exactly you develop the model, for instance, do you aim to model every detail, including the memory consumption of the program?

Page 5: “Here, we define it as the proportion of models that TLC checks without failures at least once within k generated models.”

Comment: What properties do you check and where is the non-determinism coming from?

Page 5: “Two states State1 and State2 are considered sufficiently similar, if and only if the proportion of variable values in Stateg that also exist in Stateo is greater than or equal to a threshold θ ∈ [0, 1].”

Comment: This is very ad hoc. For instance, two big states differing by a crucial boolean value might lead to completely different verification results. How do you handle the state representing the program counter then?

**Questions:**

None

---

### Official Review · Reviewer_6v73 · 2025-10-29

**Soundness:** 1
**Presentation:** 2
**Contribution:** 1
**Rating:** 2
**Confidence:** 5

**Summary:**

This paper investigates the application of Large Language Models (LLMs) to model checking. The authors propose an approach where Python programs are automatically translated into formal models that can be analyzed for correctness using assertion-based test cases. In addition, the paper introduces a benchmark suite specifically designed to evaluate and compare model-checking tools, including LLM-assisted ones. The benchmark aims to provide standardized test programs, specifications, and expected outcomes to assess accuracy and reasoning ability. The paper presents preliminary results on the feasibility of LLM-based model checking and the usefulness of the constructed benchmark for evaluating such tools.

**Strengths:**

The paper identifies a potentially impactful and under-explored direction: bridging formal verification and natural language reasoning.

**Weaknesses:**

The paper suffers from several conceptual, methodological, and presentation-related weaknesses that undermine its validity as a contribution to model-checking research:

Unsupported claims (L037–L045): The paper makes broad assertions about the state of prior work in model checking without providing supporting citations or evidence. This weakens the motivation and context for the proposed approach.

Conceptual inconsistency (L053): The authors argue that testing cannot ensure the absence of bugs but then rely on test cases to validate the correctness of their own approach. This undermines the distinction between testing and verification, as the method lacks any formal soundness guarantees.

Unclear process flow (Figure 1): The figure is never explained in sufficient detail. The transition between steps such as “Normalize” and “Remove Invalid Libraries” is opaque, leaving readers unable to follow the pipeline.

Incorrect or unjustified transformation (Figure 2): The “transformed code” does not appear semantically equivalent to the “input Python code.” For example, the termination condition and increment behavior of variable i differ between the two, suggesting that the transformation may alter program semantics. For instance, the original program terminates when i ≥ n, whereas the transformed version terminates only when pc = 3 and i ≥ n, implying termination only at a specific iteration. The paper must justify how this transformation preserves correctness.

Ambiguous metrics (L215): The meaning of reported percentages and the details of “sequential filtering” are not explained. The lack of methodological clarity prevents interpretation or replication of results.

Evaluation soundness: The evaluation relies on handcrafted models and a similarity metric to assess correctness, but no theoretical justification or proof of soundness is provided for this metric. Consequently, there is no guarantee that the generated models are correct; the oracle models themselves may be flawed.

Weak relation to verification: The evaluation uses test cases in TLC to assess model correctness, but these tests could be run directly on the program without a verifier. This makes it unclear how the proposed benchmark or methodology meaningfully supports verification tasks beyond conventional testing.

**Questions:**

How do the authors ensure that the transformed models (Figure 2) are semantically equivalent to the original Python programs?

What formal guarantees, if any, exist for the correctness of the transformation process and the similarity metric used in evaluation?

What do the filtering percentages represent, and how many stages are involved in the sequential filtering pipeline?

Given that test cases are used for validation, how does the proposed method differ from traditional testing in terms of verification guarantees?

---

### Official Review · Reviewer_7U8M · 2025-11-01

**Soundness:** 3
**Presentation:** 3
**Contribution:** 2
**Rating:** 2
**Confidence:** 4

**Summary:**

This paper presents a new benchmark set for evaluating LLM's ability to model Python programs in TLA+. The benchmark contains 400 Python programs. To obtain the ground truth, a transformation process is invoked to remove external libraries, rewrite the code to be more verification-friendly, convert the code to CFG, invoke LLMs, and manual verification. The paper evaluates several LLMs and various prompting conditions, and show that the benchmarks are very challenging for existing LLMs.

**Strengths:**

- The paper is proposing a benchmark for an important problem.
- The benchmark collection procedure the paper describes, in particular the code transformation process and the process to obtain ground truth, is sensible.
- Evaluation suggests that the benchmarks are quite challenging for existing LLMs.

**Weaknesses:**

While other parts of the workflow are described in a clear way, the paper seems to be vague about how the ground truth oracles are obtained via manual inspection and refinement. How much time does it take to examine the solution for all 400 programs manually? Who examined the programs? Is each model examined by several persons? Without these important details, it is very difficult to judge the quality of the benchmark set, as it is unclear how accurate the oracle models are.

**Questions:**

See Weaknesses

---

### Note · Authors · 2025-11-25

I have read and agree with the venue's withdrawal policy on behalf of myself and my co-authors.